# Predicting treatment response to cognitive behavior therapy in social anxiety disorder on the basis of demographics, psychiatric history, and scales: A machine learning approach

**Qasim Bukhari**[1], **David Rosenfield**[2], **Stefan G. Hofmann**[3], **John D.E. Gabrieli**[1], **Satrajit S Ghosh**[1,4]*

**1** McGovern Institute for Brain Research and Department of Brain and Cognitive Sciences, Massachusetts Institute of Technology, Massachusetts, United States of America, **2** Department of Psychological and Brain Sciences, Boston University, Boston, United States of America, **3** Department of Psychology, Philipps University Marburg, Marburg, Germany, **4** Department of Otolaryngology Head and Neck Surgery, Harvard Medical School, United States of America

* satra@mit.edu

## Abstract

Only about half of patients with social anxiety disorder (SAD) respond substantially to cognitive behavioral therapy (CBT). However, there has been little evidence available to clinicians or patients about whether any individual patient is more or less likely to have a positive response to CBT. Here, we used machine learning on data from 157 patients to examine whether individual patient responses to CBT can be predicted based on demographic information, psychiatric history, and self-reported or clinician-reported scales, subscales and questionnaires acquired prior to treatment. Machine learning models were able to explain about 26% of the variance in final treatment improvements. To assess generalizability, we evaluated multiple machine learning models using cross-validation and determined which input features were essential for prediction. While prediction accuracy was similar across models, the importance of specific features varied across models. In general, the combination of total scale score, subscale scores and responses to individual questions on a severity measure, the Liebowitz Social Anxiety Scale (LSAS), was the most informative in achieving the highest predictions that alone accounted for about 26% of the variance in treatment outcome. Demographic information, psychiatric history, personality measures, other self-reported or clinician-reported questionnaires, and clinical scales related to anxiety, depression, and quality of life provided no additional predictive power. These findings indicate that combining scaled and individual responses to LSAS questions are informative for predicting individual response to CBT in patients with SAD.

**Data availability statement:** We have made the code available. Data contains potentially sensitive patient information, including detailed demographic information which may identify individuals. Data will be made available to individual authors upon request from Jill Crittenden (jrc@mit.edu). https://mcgovern.mit.edu/tile/jill-crittenden-2/

**Funding:** QB received postdoc fellowship funding from Novartis Foundation for Biomedical Research. SG was partially supported by NIH P41 EB019936. This work was also partially supported by the Abdul Latif Jameel Clinic for Machine Learning in Health (J-Clinic) and the Poitras Center for Psychiatric Disorders Research at the McGovern Institute for Brain Research at MIT. The funding sources had no involvement in the collection, analysis and interpretation of data.

**Competing interests:** The authors have declared that no competing interests exist.

# 1 Introduction

Major psychiatric disorders have multiple treatment options available with varying levels of efficacy [1]. There is, however, limited evidence for methods of selecting optimal treatments for individual patients. For social anxiety disorder (SAD), only about half of patients exhibit substantial clinical benefits from either behavioral or pharmacological treatment [2]. The increased availability of data from clinical trials and research studies and improvements in predictive modeling provide an opportunity to evaluate whether pretreatment information can be used to select a more effective treatment for an individual diagnosed with a psychiatric disorder (i.e., personalized or precision medicine). Here we asked whether machine learning could help identify which patients with SAD would benefit from cognitive behavioral therapy (CBT) on the basis of easily available information, specifically demographic characteristics, psychiatric history, or widely used self-reported or clinician-reported rating scales.

SAD is one of the most common psychiatric conditions in the United States. According to large-scale epidemiological studies in the U.S. [3–5], SAD affects approximately one in eight people (12.1%) at some point during their lifetime, and each year 6.8% of individuals in the general population meet diagnostic criteria for the disorder. Its core symptoms [6] are a marked and persistent fear of social situations or performance, which creates intense anxiety [7]. Consequently, individuals with SAD have severe occupational, educational, and financial difficulties [8], including higher unemployment rates than patients with other affective disorders [9]. Cognitive behavioral therapy (CBT) has been used to treat SAD, and about half of patients exhibit a significant clinical response [10]. There is, however, no current way for a clinician or patient to know if a particular patient with SAD is more or less likely to benefit from CBT.

Treatment outcome prediction is an area of growing interest in computational psychiatry. Studies have used pretreatment genetics [11–13], neuroimaging, [14–18], and demographic and psychiatric data [18–26] for predicting treatment outcomes. Demographic and psychiatric data, however, are more easily collected than genetic or neuroimaging data, and it is of practical interest to understand how much they may contribute to predicting treatment efficacy. Demographic information and psychiatric history have been associated with variable treatment outcomes for many psychiatric disorders. For example, greater symptom severity has often been associated with better treatment outcome in obsessive-compulsive disorder [27]. In schizophrenia, better response to treatment has been associated with shorter disease duration [28], later age of onset of illness, more satisfactory family relationships, a more skilled occupation, and being married [19]. Mood-disorder patients without family history of mood disorder have had a better response to treatment than patients with a family history of mood disorder [12]. Socioeconomic status has also been associated with antidepressant treatment responses such that residents of middle- and high-income census tracts were more likely to respond to treatment than residents of low-income census tracts [29–31]. Similarly, another study analyzed data from a multicenter trial (N = 230) and identified pre-treatment severity of SAD as the major predictor of treatment outcome for manualized short-term psychodynamic psychotherapy (PDT), accounting for 39% of the variance [32]. In depression, pretreatment variables including history of illness, demographics and life circumstances, and baseline functioning allowed patients to be stratified into groups of low, moderate, or high risk for poor treatment response [33]. Effectiveness of internet-enabled CBT for depression and anxiety has been associated with age and severity [20].

In regard to CBT for SAD, multiple studies have reported patient characteristics associated with CBT or other behavioral therapies that were correlated with treatment response. One review summarized findings across studies as identifying earlier age of onset and greater disease severity as often correlated with lesser response, and also family history of SAD and

male gender [34]. Another review identified lower socioeconomic status and some personality traits as related to lesser response [35]. A multicenter trial (N = 230) identified pre-treatment severity of SAD as the major predictor of treatment outcome (defined by remission) for manualized short-term psychodynamic psychotherapy, but SAD severity had no significant relation to CBT response [32]. One study examined response to CBT treatment across trans-diagnostic internalizing disorders (including 23% of patients with SAD) with 324 predictors in 1020 patients, and found that predictors of greater response included younger age, personality dimensions, and lesser severity of social anxiety symptoms [36].

Most of the above studies evaluated associations (correlations) between patient variables and treatment outcomes for a specific set of patients and did not evaluate models that *predict* outcome for a new individual whose data were not used to create the model. If knowledge about a patient were to guide optimal treatment selection for a patient, the aspiration of personalized/precision medicine, that knowledge would have to be informative about new individual patients. Machine learning algorithms provide a multivariate approach to identify patterns of interest from data and create models that predict treatment in new individual patients. Indeed, machine learning algorithms have successfully predicted the response to alcohol dependence treatment and [37] to treatment for late life depression [18], remission for antidepressant symptoms [25], suicide attempts [22,26], psychiatric hospitalizations [23] and treatment response for depression and anxiety [24,25,38], and psychosis [17].

Findings using machine learning to predict treatment outcome on the basis of only demo-graphic, psychometric, and clinical data have been variable across diagnoses and outcomes. Prediction for response to CBT treatment for alcohol dependence was achieved based on demographic and psychometric assessment data [37]. Conversely, demographic and clinical data alone have not predicted treatment outcomes for response to antidepressants in patients with depression [14] or response to CBT and escitalopram in patients with SAD [11]. In both of these studies neuroimaging data were more predictive of treatment outcomes than demo-graphic and clinical data.

There are multiple kinds of machine learning algorithms, and most studies in computa-tional psychiatry have selected a single algorithm. There is, however, virtually no consensus on which different machine learning algorithms are better or worse in generating predictions of therapeutic efficacy. A few studies have compared the accuracy of different machine learning algorithms for treatment outcome for depression [38], late life depression classification and response prediction [18], prediction of clinical improvement in psychosis [17] or antidepres-sant treatment response prediction [25]. There is, however, no consensus as to which among different machine learning algorithms are better or worse in generating predictions of ther-apeutic efficacy. In regards to treatment efficacy for SAD, there has not yet been a study that compared alternative machine learning models.

In the present study, 157 SAD patients participated in 12 weekly sessions with CBT accord-ing to a standardized protocol-based group treatment [39]. All participants were part of a previously published study examining the effects of a drug (d-cycloserine) to augment CBT for SAD [40]. For the present analyses, treatment-related improvement in SAD was mea-sured by pre-post change in the Liebowitz Social Anxiety Scale (LSAS) [41], which at baseline also provided a clinical measure of symptom severity. Treatment response can be defined in several correlated ways – including *remediation* (loss of diagnosis) or *response* (degree of symptom improvement). Here we defined degree of symptom improvement as the measure of treatment response.

We first evaluated if there were any significant correlations between change in LSAS score before and after treatment and other variables in the data. Then we applied machine learn-ing algorithms to ask whether demographic and psychiatric history information of the kind

that is routinely recorded in patient visits could predict therapeutic response to CBT in SAD patients. This included age, sex, estimated age of onset, estimated duration of SAD, and psychiatric scales history for all psychiatric disorders. We next asked how much the predictions of therapeutic response improved with the addition of easily administered measures that future data collection and inference, we also asked which specific scales (for example occur primarily in research settings - clinician-administered rating scales and patient self-report surveys. Given the diversity of findings about what kinds of initial measures (age of onset, clinical severity, gender, and others) are associated with improvement, we included all available data that were collected from the patients in the study and used a data-driven approach. Finally, to optimize personality assessment, depression assessment; S1 Table in S1 Data for the full list of scales) were most relevant to the prediction. This was done in order to generate a simpler model that can achieve similar predictive accuracy (within 5%) of the full data set. These analyses were performed to identify the minimal number of measures that clinicians could collect to predict therapeutic efficacy for an individual.

The primary aim of this research was to establish if treatment response outcome can be predicted for CBT in SAD patients using readily available information. We used an out-of-sample predictive modeling approach with bootstrapped train-test splits to answer this question. As models can vary in their calculation of feature weights, we also compared the performance of a few models to better understand which features are relevant to the problem. We employed four different machine learning algorithms to examine whether one or another yielded superior protection: Lasso, Ridge Regression, Support Vector Regression (SVR) and Extra Trees. We chose these algorithms to span a range of architectural differences, linear and nonlinear approaches, and parametric complexity. Lasso uses the L1 regularization to create the sparsest models; Ridge uses L2 regularization; SVR uses support vectors for maximizing the distance between hyperplanes using linear or nonlinear kernels; and Extra Trees is a decision tree-based ensemble algorithm. We evaluated the generalizability of such predictions using cross-validation to ascertain how accurate each model was for novel, individual patients. Specifically, we estimated the prediction accuracy using data that were not used to build the predictive model and therefore did not influence the development of the model. Such an evaluation is a direct measure of how well the model will perform on data from new individuals. Since machine learning models can use complex linear and nonlinear relations across information, we also examined which specific features (features are variables that are inputs to the machine learning models) were considered most informative or influential in each model to determine.

## 2  Methods

The data set has been described in detail [42]. The data was accessed on September 1, 2021 for research purposes of this study and the authors had no information that could identify individual participants. Participants (91 men and 66 women) had a mean age of 32.4 years (range 18–66 years). The mean estimated age of onset of SAD was 12.3 years. Participants were recruited from three recruitment sites: the Center for Anxiety and Related Disorders at Boston University (n = 69), the Center for Anxiety and Traumatic Stress at the Massachusetts General Hospital (n = 51), and SMU (Southern Methodist University) (n = 37). Patients gave written informed consent to all procedures, which were approved by the internal review boards of all three sites. Data from all 157 patients with SAD (generalized subtype) who completed the CBT study were included in these analyses.

Patients were not taking concurrent psychotropic medication for at least two weeks prior to CBT initiation. Consistent with an earlier study involving both clinical sites [42], diagnoses were confirmed at the Massachusetts General Hospital with the Structured Clinical Interview

for *DSM-IV [43]* or at the Center for Anxiety and Related Disorders with the Anxiety Disorders Interview Schedule for *DSM-IV [44]*. Severity of social anxiety was measured using the clinician-administered version of the LSAS [41]. Exclusion from the study occurred in the case of a lifetime history of bipolar disorder, schizophrenia, psychosis, delusional disorders or obsessive-compulsive disorder; an eating disorder in the past 6 months; and a history of substance or alcohol abuse or dependence (other than nicotine) in the last 6 months or post-traumatic stress disorder within the past 6 months or neurological disorders or other serious medical illnesses.

Clinical assessments took place both immediately prior to and at the end of the 12 group-based CBT treatment sessions (13th week). For the present analyses, we used the change in the total LSAS value between the pretreatment baseline and the post-treatment assessment as the outcome measure. This variable offered a continuous and fine-grained measure of patient response to treatment and allowed us to compare results to neuroimaging studies [15,16] that used similar LSAS scores as the outcome measure.

Additional information collected from the patients included demographic information (e.g., age and sex, Table 1), psychiatric history (e.g., duration and severity of SAD, comorbid diagnoses, Table 1), Liebowitz Social Anxiety Scale (LSAS) [41], Social Phobic Disorders Severity Change Form (SPDSC) [45], LIFE-RIFT (Range of Impaired Functioning Tool) questionnaire, Social Phobia and Anxiety Inventory (SPAI) [46], Quality of Life Enjoyment and Satisfaction Questionnaire (QLESQ) [47], Montgomery–Åsberg Depression Rating Scale (MADRS) [48], Pittsburgh Sleep Quality Index (PSQI) [49] and the NEO Personality Inventory (NEO) [50]. For analyses, we included all summary scores and all responses to individual questions of these scales (S1 Table in S1 Data). The complete list of variables used in the analysis is given in S1 Tablein S1 Data.

## Data preprocessing

We employed the 'most frequent' data imputation strategy for missing data. Any variable that had less than 10% missing values were imputed. In our data, either the variables had less than 10% missing for different subjects, or greater than 50% missing. For any variable missing greater than 50% of the values we excluded that variable from the study. The data had 10% missing values across different variables. We also standardized the features by removing the mean and scaling to unit variance. Both of these operations were performed within training datasets and applied to test datasets to maintain independence of training and test data.

## Model development

We compared the performance of several machine learning models of increasing complexity, Lasso, Ridge Regression, Support Vector Regression (SVR) and Extra Trees. We used hyper-parameter tuning for all the algorithms in a nested cross-validation approach. In nested cross validation we treated model hyperparameter optimization as part of the model fitting process. The training data during the fitting process was used to tune model parameters. The model parameters were optimized using a grid search procedure which sweeps through the parameter space of the model to find the best performing model parameters using the results of a 5-fold validation.

The tuning range for SVM included the following parameters: linear and rbf kernel, C from 0.001 to 10 with an interval of 10, gamma from 0.001 to 1 with an interval of 10. The tuning range for Ridge Regression included the following: alpha from 0.1 to 10 with a step of 0.1, fit intercept, normalize and solver with selection between lsqr, auto, svd. Lasso was used with fit intercept and normalized. Extra Trees algorithm was used with 100 estimators. Finally

**Table 1. Demographics and psychiatric history of the sample.**

*Demographic and Clinical Characteristics of the Sample*

| Sample characterstics | | N | % |
|---|---|---|---|
| Gender | | | |
| | Male | 91 | 57.96 |
| | Female | 66 | 42.04 |
| Race | | | |
| | Native American/Alaska Native | 1 | 0.64 |
| | Black or African American | 15 | 9.55 |
| | Native Hawaiian or Other Pacific Islander | 1 | 0.64 |
| | White | 113 | 71.97 |
| | Asian | 20 | 12.74 |
| | Other | 7 | 4.46 |
| Ethnicity | | | |
| | Hispanic/Latino | 17 | 10.83 |
| | non Hispanic/Latino | 140 | 89.17 |
| Education | | | |
| | Partial high school | 66 | 42.04 |
| | Partial college | 43 | 27.39 |
| | High school graduate | 9 | 5.73 |
| | Graduate school | 37 | 23.57 |
| | College graduate | 2 | 1.27 |
| Occupation | | | |
| | Administrative | 23 | 14.65 |
| | Clerical | 14 | 8.92 |
| | Executive | 9 | 5.73 |
| | Manager/Professional | 61 | 38.85 |
| | Never worked | 5 | 3.18 |
| | Semi skilled | 10 | 6.37 |
| | Skilled | 21 | 13.38 |
| | Unskilled | 14 | 8.92 |
| Marital status | | | |
| | Divorced | 7 | 4.46 |
| | Living with partner | 12 | 7.64 |
| | Married | 39 | 24.84 |
| | Separated | 1 | 0.64 |
| | Single | 97 | 61.78 |
| | Widowed | 1 | 0.64 |
| Lifetime diagnoses (*DSM–IV-TR*) | | | |
| | Major depressive disorder | 75 | 47.77 |
| | Bipolar disorder | 2 | 1.27 |
| | Anorexia | 1 | 0.64 |
| | Post-traumatic stress disorder | 10 | 6.37 |
| | Panic disorder | 9 | 5.73 |
| | Obsessive-compulsive disorder | 2 | 1.27 |
| | Alcohol abuse or dependence | 9 | 5.73 |
| | Specific phobia | 11 | 7.01 |
| | | M(mean) | SD |
| Age (in years) | | 32.42 | 10.13 |

cross-validated estimators from all the models were used for all the analysis. The analysis was done using pydra-ml [51], based on scikit-learn, and the scripts are provided on GitHub at https://github.com/qasim85/SAD_behav_manuscript. Machine learning models were trained on baseline features, that is, information available before the start of treatment to predict the change in LSAS as a result of the intervention.

## Model validation

We employed stratified shuffle split randomized sampling, i.e., repeated sampling cross-validation with 1000 iterations, to ascertain generalizability of the model (sensitivity and specificity) to unseen data and to ascertain a distribution of model performance, while removing order effects. Given the size of the dataset, generating a pure holdout test set would have resulted in sampling a limited amount of phenotypic and clinical variation. Instead, we chose to use the bootstrapping procedure to estimate models on several subsets of the data to provide a distributional view of the performance of the model. While we did not use a completely held out dataset, we ensured that there was no data leakage and none of the data used in model building was used in the model validation. For each iteration, we created a pair of training and testing data by randomly sampling and assigning 80% of patients to the training set (for creating the model) and assigning the remaining 20% of patients to a test set (for evaluating the model). Thus, in each pair there was no participant common to both the training and test set. In each iteration, the performance of the trained model was evaluated using variance explained ($R^2$) of the LSAS change in the test set. Thus, the distribution of model performance on the test set of each of the 1000 iterations provides an estimate of the generalizability of the model to new samples. The shape of the performance distribution can provide clues to sampling biases in the data. Furthermore, we tested for significance using a permutation testing approach [52] that compares the mean performance of the model to the performance distribution of a corresponding null model. The null model was created by training the same algorithms on permuted training data. For each of the 1000 iterations, this permuted data is created by randomly permuting the scores associated with each sample in the training data.

Fig 1 shows the workflow of the analysis, including the grid search for finding the best parameters within the shuffle split strategy used for validation and evaluating the model on test data. The model is also compared with a null model that uses permuted labels, to determine performance above chance prediction.

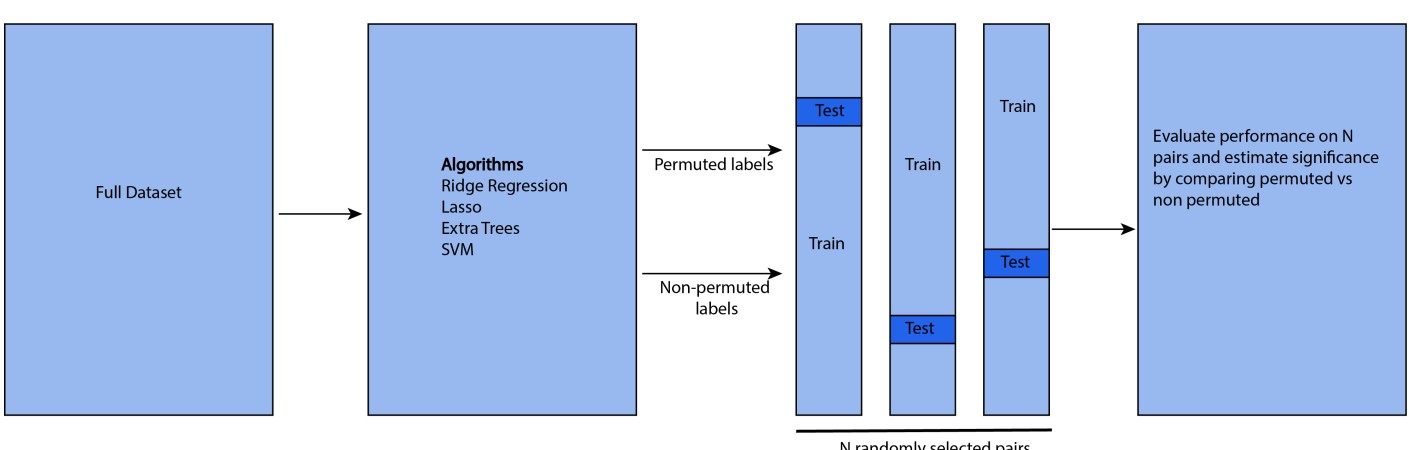

**Fig 1. Workflow of the analysis pipeline.**

## Hypothesis testing

First, we tested the hypothesis that the performances of the four models trained with all measured variables (demographic information, psychiatric history, and scales) are better than chance. Second, in order to evaluate the usefulness of the scales, we tested the hypothesis that the performance of the model trained with all measured variables is better than the model using only demographic information and psychiatric scales.

## Feature importance using SHAP values

Machine learning interpretability is important because it allows machine learning to not be a black box of inputs and outputs, but rather provides insights as to the features that are most important for the predictions. Feature importance helps in estimating how much each feature of the data contributed to the model's prediction. We used SHAP [53] value, to assess the feature impact on model output. SHAP is a game theoretic approach that provides a unified approach for interpreting output of machine learning methods. SHAP computes the feature importance based on magnitude of feature attributions and computes the impact of features by combining feature importance with feature effects. SHAP provides an advantage over using coefficients of each model to explain the importance of the models, since many models, for example SVMs with nonlinear kernels, do not have a direct way of providing coefficients to assess the important features. Furthermore, SHAP, as a training algorithm agnostic approach, was used to compute feature impact across the different machine learning models and it also allowed us to directly compare feature consistency and differences across models. We also computed an aggregate feature importance measure by weighting the SHAP values of each algorithm by the performance of the algorithm. This allowed penalizing weaker algorithms from contributing heavily to feature importance. Due to differences between algorithms, chosen features can vary because of intrinsic algorithmic details (e.g., type of regularization). Therefore, to provide a more comprehensive view of the relations across features, we estimated the distance correlation of all features using Python library *dcor*, and grouped them according to the categories of questions to identify which scales are related to each other.

## Optimization analysis to identify the most relevant scales

In the present study, multiple questions as part of different scales (e.g., LSAS, MADRS, NEO) were administered that took over 2 hours to complete. To optimize future data collection, we aimed to determine the smallest combination of scales that provides predictive value comparable to the entire data collection. Using the same nested cross validation procedure outlined before, we evaluated the predictive performance of our models on all com binations of scales used in this study. For each combination of scales (N = 258; see S2 Table in S1 Data), the measured item values and the computed total scores were used as input features for training the models. These performances were ranked to find the smallest combination of scales that were closest to the performance of the models using all measured variables.

## Results

First we examined correlations between pretreatment measures and response to CBT (change in LSAS scores). Then, we examined how well the four machine learning models predicted individual patient responses to CBT (1) on the basis of demographic and psychiatric scales history information, (2) on the basis of information from the scales and (3) on the basis of all information (demographic history, psychiatric history, and questionnaires). We further investigated the machine learning results by identifying the important variables (features) that

contributed most to the machine learning prediction models. Finally, we examined what were the best variables for predicting response to CBT so that such variables could be collected most efficiently.

## Explained variance and correlations between pre-treatment scores on scales and subscales and LSAS change scores

Explained variance and Pearson correlations between LSAS change at the end of treatment and assessments prior to intervention varied across the scales (Table 2), with some summary scores showing strong and significant effect sizes. In all cases, worse or higher scores at baseline correlated with greater improvement.

## Explained variance and correlations between all pre-treatment individual variables and LSAS change scores

Table 3 shows the top 10 significant correlations between LSAS change and all baseline variables including demographic information, psychiatric history and scales. The strongest associations all came from the scales at the levels of total scores, sub-scores, or individual questions.

**Table 2. Explained variance and Pearson correlations between pre-treatment summary scores and LSAS change at the end of treatment.**

| Clinical rating scale | Explained Variance | Pearson Correlation |
|---|---|---|
| Liebowitz Social Anxiety Scale (LSAS) | 0.33 | 0.33 |
| LSAS fear subscale | 0.31 | 0.31 |
| LSAS avoidance subscale | 0.27 | 0.27 |
| LIFE-RIFT (Range of Impaired Functioning Tool) questionnaire | 0.09 | 0.09 |
| Social Phobia and Anxiety Inventory (SPAI) | 0.05 | 0.05 |
| Quality of Life Enjoyment and Satisfaction Questionnaire (QLESQ) | 0.00 | 0.00 |
| Montgomery–Åsberg Depression Rating Scale (MADRS) | 0.00 | 0.00 |
| Pittsburgh Sleep Quality Index (PSQI) | 0.00 | 0.00 |

*For bivariate correlations explained variance is the same as Pearson correlation.

**Table 3. Top 10 explained variance and correlations between all pre-treatment individual variables, and LSAS change scores.**

| Question | Explained Variance | Correlations |
|---|---|---|
| Liebowitz Social Anxiety Scale (LSAS) | 0.33 | 0.33 |
| LSAS fear subscale | 0.31 | 0.31 |
| LSAS avoidance subscale | 0.27 | 0.27 |
| LSAS fear question 4 | 0.16 | 0.16 |
| Social Phobia and Anxiety Inventory question 36 | 0.13 | 0.13 |
| LSAS avoidance question 1 | 0.12 | 0.12 |
| LSAS fear question 3 | 0.11 | 0.11 |
| LSAS avoidance question 4 | 0.10 | 0.10 |
| Social Phobia and Anxiety Inventory question 24c | 0.10 | 0.10 |
| Social Phobia and Anxiety Inventory question 20d | 0.10 | 0.10 |

*For bivariate correlation, explained variance is the same as Pearson correlation.

## Machine learning predictions for individual treatment outcome

**Models with only demographic information and psychiatric history.** Models including pre-treatment demographic and psychiatric history information (e.g., age, sex, estimated age of onset, estimated duration of SAD, and psychiatric history for all psychiatric disorders prior to start of treatment;complete list in S1 Table in S1 Data) accounted for minimal amounts (0% to 11.5%) of the variance in the predicted treatment outcome (Table 4; Fig 2).

**Models with only scales.** Models including only pre-treatment scales accounted for up to 26% of the variance in therapeutic outcome (Table 4).

**Models with all variables.** Models that included all pre-treatment variables consisting of demographic information, psychiatric history, and scales as input accounted for up to 26% of the variance in therapeutic outcome (Table 4; Fig 2). Predictive performances varied across models. Lasso was the best performing model with more than 26% of explained variance for predicting treatment response. Support vector regression showed the lowest performance across all models. Table 4 also used the median explained variance with a 90% interval bound for each algorithm.

**Table 4. Median explained variance to predict therapeutic outcome (LSAS change) for different machine learning algorithms using only demographic information and psychiatric history, only scales, or all of these variables. The 90% interval for each model is reported alongside the median value.**

| Machine Learning Algorithm | Median Variance Explained (in percentage) | | |
|---|---|---|---|
| | Demographics and clinical history | All scales | All measured variables |
| Ridge Regression | 4.32 [-21.20 – 18.70] | 24.32 [-11.81 - 37.94] | 26.71 [3.28- 38.51] |
| Lasso | 11.03 [-2.89 – 19.75] | 26.16 [2.31 - 37.88] | 24.06 [0.43- 36.82] |
| Extra Trees | 1.06 [-59.46 – 31.58] | 24.29 [-22.14 - 43.15] | 21.40 [-26.81- 41.38] |
| Support Vector Regression | 0.00 [-3.00 – 3.00] | 16.14 [-14.45 - 23.45] | 15.80 [-20.95- 23.28] |

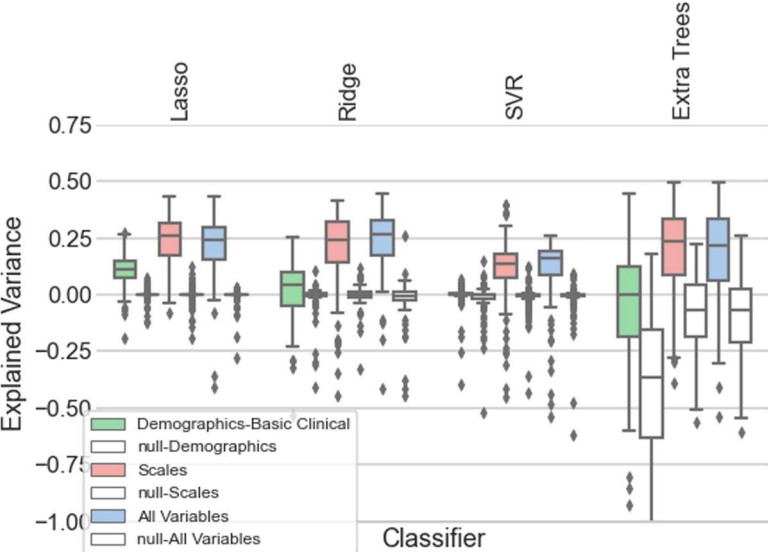

**Fig 2. Machine learning prediction results of LSAS change scores using only demographics and psychiatric history, only scales or all measured variables.**

Fig 2 shows the explained variance box plots for different machine learning algorithms (Lasso, Ridge Regression and Support Vector Regression and Extra Trees) using only demographics and psychiatric history, only scales or all measured variables using LSAS change at the end of the treatment. All explained variances were compared against a null model that was tested with shuffled labels. Box plots show the interquartile range thus we have range in 5, 25, 50, 75, 95 percentiles.

**Significance and null hypothesis testing - models with only demographics and psychiatric history information or only scales or all variables.** Both models with inputs from demographic information and psychiatric history and models with inputs from scales predicted improvement significantly for Lasso (p = 0.001), Extra trees (p = 0.001) and Ridge (p = 0.008), with only SVR not significantly different from the null hypothesis. The performance of the prediction models including all variables was significantly better (p = 0.001) than the models comprising only demographic information and clinical information. This significant difference was observed for all four models. Moreover, the performance of the prediction models including inputs from scales was significantly better than the models including demographic information and psychiatric history, for Lasso (p = 0.008), Extra trees (p = 0.02), Ridge (p = 0.03), and SVR (0.03). Models with inputs from scales were not significantly more accurate than models with all variables.

**Model features using SHAP.** Our model identified different sets of important features for each of the machine learning algorithms. Using SHAP, which provides a unified measure of feature importance, we determined that the most important model features varied in their ranking for each of the machine learning algorithms used for predicting the treatment response at the immediate end of the treatment. In most models, responses to individual LSAS questions were ranked highly, but the exact ranking of these items varied across the classes of models. S2 Table in S1 Data shows all the most important features identified by SHAP across different machine learning algorithms ranked according to their weighted mean. The distance correlation across all the features are grouped according to the categories of questions, representing the relationship between features, is shown in S1 Fig in S1 Data.

**Optimization of the most relevant scales through exhaustive search.** The LSAS total scale, two subscales, and individual questions were the best predictors of treatment outcome as evaluated using different machine learning algorithms on an exhaustive 258 different combinations of scales in the data. Fig 3 shows the distribution of out of sample performance on assessment combinations that included or discarded the LSAS subscale. Table containing combinations of feature subsets and their corresponding explained variance using Extra Trees is given as S3 Table in S1 Data. The graphical representation of this table is shown in Fig 3

The above results identified that explained variance was driven mostly by LSAS values. Therefore, we further investigated the LSAS features only to narrow down the important features among LSAS questions. S4 Table in S1 Data shows all the most important features identified by SHAP across different machine learning algorithms ranked according to their weighted mean using only LSAS items level scores as input. The important features vary across different models but top four features according to the weighted mean are within the top 10 important across the models.

## Discussion

The major goal of these analyses was to discover how well machine learning could predict, from initial or pre-treatment information, variance in response to CBT among patients with SAD, and to do so with low-cost and easily available demographic information, psychiatric history, and self-reported and clinician-reported scales. We found that up to 26% of the

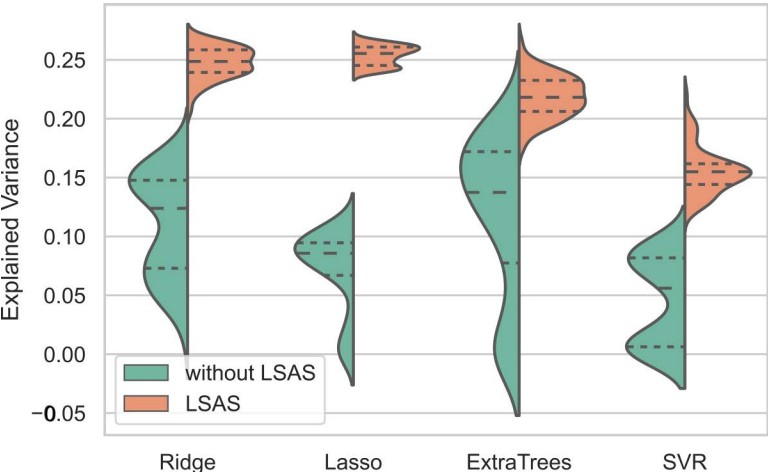

**Fig 3. Explained variance of features subsets using different machine learning algorithms: Fig 3 shows all the possible combinations of feature subsets, according to S3 Table** in S1 Data**, with corresponding explained variance estimated using different machine learning algorithms (Lasso, Ridge Regression, Support Vector Regression (SVR) and Extra Trees).** The y-axis shows the explained variance for feature combinations with and without LSAS questionnaire.

variance could be predicted based on such information. Critically, this prediction was based on repeatedly training machine learning models on a random group of patients, and testing those models on a different random group of patients with no overlap between the training and testing participants in each iteration. If predictive models were to be used in some way to guide clinical decision-making, these models would have to operate at the single patient level and be applicable to a new patient [11].

Pre-treatment severity, as measured by the LSAS, accounted for all of the variance in response that could be predicted from the initial measures. Other pre-treatment demographic, psychiatric history, comorbidity, and questionnaire measures added very little predictive information beyond the initial LSAS values. The combination of the LSAS summary score, subscales, and individual questions was a better pre-treatment predictor of response to CBT than the single LSAS summary score. Researchers often focus on total or summary scores, or sometimes subscales, but several machine learning studies have found that specific items on these scales make additional useful contributions to predictions [21,54]. Although LSAS total, subscale, and individual questions are somewhat redundant (as the individual questions are summed to yield summary and subscale values), inclusion of all three types of LSAS scores improved prediction accuracy.

Clinicians and researchers typically use overall summary scores from self-report questionnaires to evaluate diagnoses or response to treatment. Indeed, such summary scores typically have the most reliability and validity relative to subscores or individual items. In the present study, however, prediction was most accurate when data from individual items, subscores, and the total score were all included. A similar benefit of including all data from a single clinical questionnaire for ML prediction was observed for a study predicting the development of bipolar disorder in children over a 10-year period [55]. At a practical level, clinicians and researchers would not be expected to implement ML algorithms to calculate likelihood of treatment efficacy (or development of a diagnosis), but rather risk calculators would need to be made available that would automatically compute such probabilistic outcomes. Such risk calculators are already in use in other areas of medicine, but not yet for psychiatric diagnoses or treatments.

The finding that more severe pre-treatment symptoms predicted greater response differs from prior studies of SAD reporting that more severe symptoms were associated with lesser response to CBT [34,36]. Consistent with the present study, however, meta-analysis with data from 12 trials with 1246 SAD patients revealed that CBT led to significant symptom reduction, and individuals with higher baseline SAD severity experienced greater symptom improvement [56]. It is unclear if these differences relate to variation in patient populations, specific severity measures, or CBT delivery.

There are multiple kinds of machine learning models and no principled way to determine which one is most useful to predict individual response to mental health treatments in general or to CBT for SAD in particular. We took an empirical approach and examined the predictive accuracies of four different machine learning models Lasso, Ridge Regression, Support Vector Regression and Extra Trees. Lasso yielded the highest level of predictive accuracy, followed closely by Extra Trees. Previous studies comparing multiple machine learning algorithms did not find a consensus as to which algorithm performed better or worse in making predictions [18,25,38,17]. Future studies ought to compare models and determine whether some kinds of models are generally superior or whether different kinds of models are superior for different kinds of treatments (e.g., behavioral or pharmacological) or different diagnoses (e.g., anxiety or depression).

Furthermore, by reporting only mean performance, the variability inherent in such models when predicting information about individuals in a population has not previously been conveyed. Therefore in our work, we used a bootstrapped cross validation method that generated a distribution of performance metrics in held out samples rather than a single value. Thus, a confidence interval of the performance metrics could be obtained, which provides a more nuanced view of the performance of the model when large test datasets are not available.

A question of interest is what specific kinds of information best predicted response to CBT in individual SAD patients among the 369 pretreatment variables that were examined. A concern about machine learning is that it can function as a "black box" transforming many inputs into predictions without clarity as to what inputs are most informative, potentially incurring bias inherent in many datasets that are unrelated to the clinical intent. We addressed this concern by using SHAP [53] to identify which inputs were most informative for the models. We found that different machine learning algorithms varied as to which inputs were most informative. This is expected given the different optimization approaches these algorithms use. For example, Lasso will disregard any feature that is closely correlated with another feature. To further evaluate which input questionnaires were most predictive of response to treatment, we tested all possible combinations of these questionnaires' item responses and scores. This approach identified the most important subset of questionnaires relevant to this dataset. The SHAP method identified that several components of the LSAS questionnaire were important along with other features for predicting outcome. This post hoc analysis demonstrated that LSAS summary score, subscale scores and individual questions accounted for most of the explained variance across all the machine learning algorithms. It is worth noting that because the treatment response was defined for each patient as the reduction in SAD severity defined by the total score on the LSAS, pre-treatment LSAS values may have been favored as predictors of outcome.

Many studies report only correlations between pre-treatment information and treatment response, but nearly all these studies report such findings from a specific set of patients without evidence that the correlations will generalize to another set of patients. In contrast, we tested the models trained in one set of patients in another out-of-sample set of patients to measure the generalizability of findings. Some pre-treatment measures that correlated with treatment response exhibited little generalizability across samples. For example, pre-treatment

scores on the LIFE-RIFT ($r = 0.30$, $p < .001$) and SPAI ($r = 0.22$, $p < 0.005$) correlated positively with treatment response, but neither were among the top 20 features included in the SHAP analysis. When we did an analysis of these two scales (LIFE-RIFT and SPAI), including individual questions, performance was also low in out-of-sample prediction [S2 Table in S1 Data, Fig 3]. These examples illustrate that correlations can have very limited value in relation to generalizable models that predict treatment response across patients.

Similar to our findings, most machine-learning studies using readily available pre-treatment demographic information, psychiatric history, and scale information to predict treatment response have accounted for about a quarter to a third of individual variance in patient outcomes [54]. Hornstein et al. reported a balanced accuracy of 60% for response to digital mental health intervention treatment outcome prediction in anxiety and depression [24]. This study was somewhat different than most other studies in that participants were not clinically diagnosed.

It is unclear why so many other measures did not contribute to prediction beyond the LSAS. The LSAS directly measures the symptoms of SAD, and clearly has the most face validity when considering response which is defined as a reduction of symptoms. Perhaps this is why symptom severity has been the most consistent predictor of response to CBT across studies [34,36,56].

The present finding that more severe pre-treatment symptoms predicted greater response differs from prior studies of SAD reporting that more severe symptoms were associated with lesser response to CBT [34,36]. Consistent with the present study, however, meta-analysis with data from 12 trials with 1246 SAD patients revealed that CBT led to significant symptom reduction, and individuals with higher baseline SAD severity experienced greater symptom improvement [56]. It is unclear if these differences relate to variation in patient populations, specific severity measures, or CBT delivery.

Neuroimaging studies, on the other hand, have accounted for substantially more of such variability. About a third of the SAD patients in the current report underwent magnetic resonance imaging, and such neuroimaging accounted for up to 60% of the variance in response to CBT in patients with SAD [15,16], but in a cohort about a quarter of the size of this sample. Another study with SAD patients reported that functional MRI data predicted response vs. nonresponse to CBT plus escitalopram, but that patient history and questionnaires did not improve prediction accuracy [11]. Similarly, a direct comparison of neuroimaging versus demographic/questionnaire information found that pre-treatment neuroimaging accounted for much more of the individual variance among patients with depression receiving sertraline than did demographic/questionnaire information, which was not above chance[14]. Another study identified that dorsal anterior cingulate cortex (dACC) activation to aversive faces predicted treatment response to selective serotonin reuptake inhibitors (SSRIs) combined with cognitive-behavioral therapy (CBT), achieving 81% accuracy of response to treatment in individual-level predictions for SAD [57]. While neuroimaging appears to offer considerably more predictive accuracy, most of these studies are based on limited datasets and limited independent validation.

We found that the ML analyses only explained 26% of the variance in treatment response. Neuroimaging of a small subset of these patients, in combination with initial LSAS scores, explained much more of response. It is possible that the intrinsic variability of self-reported measures and the lack of biological measures (brain, genetic) and exposome measures (lifetime exposures to events prior to and during treatment) contributed to the limited variance accounted for. Analysis of self-report and clinician-report variables is the practical approach to treatment response outcome prediction without relying on hard to obtain neuroimaging data, and in a situation where other predictors simply do not exist.

Several limitations of the present study may be noted. One of the limitations of this study was the reliance on data from a clinical trial that has a limited sample size (157 patients) in relation to the many individual and environmental factors that might influence outcome. Another limitation is the lack of a group receiving an alternative treatment, such as pharmaceutical treatment. The practical question for the patient and physician is to select among alternative treatments rather than whether to treat or not treat.

Our data has 10% missing values across different variables. One of the challenges in selecting the best imputation strategy was that we did not know *a priori* if there exists dependencies between certain variables. All the missing variables were from the categorical variables and therefore mode imputation is most suited for such kind of missing values. When data were restricted to a subset of participants who had no missing data, the analyses (not included) yielded near identical results and the same conclusions.

Lack of a comparison group is a limitation of this dataset and this study. A comparison treatment, such as a pharmacological treatment, might help reveal how alternative treatments should be weighed for a patient. This would reflect the practical clinical need to offer the treatment most likely to help a patient, even if that treatment has a modest likelihood of being helpful. The present study supports the conclusion that including LSAS scores should be a component of such a process. Thus, these results needed to be considered with great care, since a patient with poor treatment response prediction may still have a better outcome than responses from alternative treatments. Therefore, any results from this study may not be decisive in treatment selection.

There is a compelling motivation to develop machine-learning or other predictors of response so that personalized or precision computational psychiatry can guide patients to effective treatments rather than the 50-60% treatment failure rate that patients must now endure. It is as yet unclear what level of predictive accuracy could be useful in treatment selection, although at present such selection is essentially random from a scientific perspective. One approach to this question has been to define a criterion of treatment efficacy, such as percent reduction in symptoms or remission. Any criterion short of remission is somewhat arbitrary, but potentially useful. For example, when setting a criterion of a 50% reduction in LSAS scores for SAD patients receiving CBT, pre-treatment neuroimaging yielded 81% accuracy, 84% sensitivity and 78% specificity in identifying individual patient outcomes[15]. It will be important to develop useful metrics of prediction that can be employed by clinicians to help patients receive effective treatments sooner and rationally rather than later and randomly as is the current practice. Therefore, our work provides a quantitative approach to predict response to treatment in social anxiety disorder using only readily available clinical information that can help clinicians guide in treatment selection.

In conclusion, these findings do point to the value of pre-treatment LSAS scores for predicting, to a moderate degree, treatment response to CBT in SAD patients. The LSAS score is easy to acquire, and it may be a useful element in the development of a practical risk calculator for assessing the probability of individual-specific benefits of such treatment.

## Supporting information

**S1 Data.**  S1 Table. List of all features grouped in to their respective subsets. Table containing all the features used in the analysis and their groups into respective subsets are given in the excel file. S2 Table. Weighted mean rank of features from all scales, subscales and questions derived from SHAP values. S3 Table. Explained variance of features subsets. S4 Table. Weighted mean rank of features from LSAS scales, subscales and questions derived from SHAP values. S1 Fig. Distance correlations of all features grouped according to categories. S1

Fig in S1 Data shows the relationship across all features by estimating the distance correlations and grouping them by the categories of questions. This represents the relation between various scales.
(ZIP)

## Author contributions

**Conceptualization:** Qasim Bukhari, Stefan G. Hofmann, John D.E. Gabrieli, Satrajit S Ghosh.

**Data curation:** Qasim Bukhari, David Rosenfield.

**Formal analysis:** Qasim Bukhari.

**Funding acquisition:** John D.E. Gabrieli.

**Methodology:** Qasim Bukhari.

**Software:** Qasim Bukhari.

**Supervision:** Stefan G. Hofmann, John D.E. Gabrieli, Satrajit S Ghosh.

**Visualization:** Qasim Bukhari.

**Writing – original draft:** Qasim Bukhari.

**Writing – review & editing:** Stefan G. Hofmann, John D.E. Gabrieli, Satrajit S Ghosh.

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
