## [Decision Letter · Decision Letter 0]

16 Jan 2024

PONE-D-23-36553Predicting Treatment Response to Cognitive Behavior Therapy in Social Anxiety Disorder on the Basis of Demographics, Psychiatric History, and Scales: A Machine Learning ApproachPLOS ONE

Dear Dr. Bukhari,

Thank you for submitting your manuscript to PLOS ONE. After careful consideration, we feel that it has merit but does not fully meet PLOS ONE’s publication criteria as it currently stands. Therefore, we invite you to submit a revised version of the manuscript that addresses the points raised during the review process.

Please respond to all reviewer comments. 

We look forward to receiving your revised manuscript.

Kind regards,

Kymberly D. Young, Ph.D.

Academic Editor

PLOS ONE

“QB received postdoc fellowship funding from Novartis Foundation for Biomedical Research as well as Abdul Lateef Jameel Clinic for Healthcare at MIT.”

6. PLOS requires an ORCID iD for the corresponding author in Editorial Manager on papers submitted after December 6th, 2016. Please ensure that you have an ORCID iD and that it is validated in Editorial Manager. To do this, go to ‘Update my Information’ (in the upper left-hand corner of the main menu), and click on the Fetch/Validate link next to the ORCID field. This will take you to the ORCID site and allow you to create a new iD or authenticate a pre-existing iD in Editorial Manager. Please see the following video for instructions on linking an ORCID iD to your Editorial Manager account: https://www.youtube.com/watch?v=_xcclfuvtxQ.

Reviewers' comments:

Reviewer's Responses to Questions

**Comments to the Author**

1. Is the manuscript technically sound, and do the data support the conclusions?

Reviewer #1: Yes

Reviewer #2: No

2. Has the statistical analysis been performed appropriately and rigorously? 

Reviewer #1: Yes

Reviewer #2: No

3. Have the authors made all data underlying the findings in their manuscript fully available?

Reviewer #1: Yes

Reviewer #2: No

4. Is the manuscript presented in an intelligible fashion and written in standard English?

Reviewer #1: Yes

Reviewer #2: Yes

5. Review Comments to the Author

Reviewer #1: This paper investigates predictors of treatment outcome in cognitive behavioral therapy for social anxiety disorder. Potential predictors in focus are demographic factors, psychiatric history and various self-rated scales. The most predictive elements were found to be different summary scores and items on the LSAS.

The paper is interesting and well-written but it is also afflicted with a concern that needs to be addressed.

Comments:

1.This research area, prediction of treatment outcome in mental health conditions, is filled with conflicting findings and few conclusions that can be drawn despite the fact that there are plenty of studies out there. One of the reasons for this may be that researchers tend to insert too many predictors into their models, without considering from a theoretical viewpoint which predictors should be of interest to look into. This leads to chance findings and heterogenous results. I understand the potential point with the current data-driven approach, but I would like for the authors to add a paragraph in the Introduction where they discuss why the baseline predictors that they are investigating may be associated with a better or worse treatment outcome. Why could for instance initial severity of social anxiety symtoms affect the outcome? Or level of depression, sleep quality, age or education level? Based on what we know about social anxiety disorder and cognitive behavioral therapy. And the discussion should address why most predictor variables did not contribute to explain the variance in treatment outcome. What does this potentially say about CBT for social anxiety disorder?

2. Only 26% of the variance was found to be explained by the 300+ predictors included in the models. What can be speculated about the 74% unexplained variance? Which are the omitted predictors?

3. Should the conclusion be that LSAS at baseline predicts LSAS at post-treatment?? It would be helpful with a concluding paragraph at the end of the discussion.

Reviewer #2: The authors examined the performance of 4 different modeling frameworks (ridge, lasso, support-vector regression, and extra trees) in predicting change in symptoms using up to 369 features from 157 patients who received a 12-week group CBT intervention. They observed that LASSO did the best in most comparisons and that baseline features of the symptom measure itself served as the strongest indicators of pre-post change. The best identified model explained about ~26% of the variance in the authors’ cross-validation procedure.

Developing prediction models to identify which patients are likely to respond well to particular treatments for social anxiety disorder is a laudable goal – and this paper has notable strengths. The analytic steps were generally well considered and well described, the sample was well characterized, and the manuscript was generally well written.

Some concerns dampen enthusiasm for the work in its current form:

Primary concern:

The paper seems to be aiming at two primary goals: developing a predictive algorithm and comparing different modeling algorithms. This split focus detracts, and it seems to have hindered the authors in achieving either goal. For example, in the development of the predictive algorithm, the authors only report on R-squared as the outcome. Perhaps more important for evaluating the utility of the model is information about how well (or poorly) calibrated the individual point predictions of the model are in the cross-validated sample. That is, if a clinician had access to the variables described in, e.g., table 2, what could they expect for a response profile and how confident could they be in that estimate? Regarding the second aim, it’s not clear that this data can resolve the question of which of the 4 modeling approaches is best. The sample is quite small for this sort of approach, particularly given the large number of features. There is a belief that ML techniques can accommodate large numbers of features even in small samples, but empirical tests of that hypothesis suggest it unlikely to be true (see Riley 2021 J. Clinical Epidemiology, Riley el at 2020 BMJ). Furthermore, it’s not quite clear how the results of any such comparison in a study like this wouldn’t simply be seen as sample- and context-specific, without a series of simulations to suss-out more general features that could help future researchers determine which approach is likely to be better for their sample and context. Furthermore, there have been advancements in ensemble-based ML methods that attempt to sidestep entirely needing to choose the one “best” algorithm for a particular application.

Additional concerns:

-Given the sophistication of the rest of the analyses, I was surprised by the choice of imputation strategy (mode imputation) and the lack of detail provided regarding missing data. How much data needed to be imputed, both for the primary outcome and each of the IVs? Although straightforward, mode imputation does not preserve any of the relationships between the variables. Additional detail and justification is required.

-Additional discussion of the limitations surrounding the lack of a comparison group are warranted. The fact that baseline values were the strongest predictor suggests the real possibility of regression-to-the-mean confounding the results. Moreover, without a comparison, care needs to be taken in characterizing how such models can be used. For example, the model may predict that Patient X will fare poorly in this treatment, but it is possible that they would fare even worse in other treatments.

-In the introduction, the authors claim that “demographic and clinical data alone have not predicted treatment outcomes for response to antidepressants in patients with depression”. That is not an accurate reflection of the state of the literature. See e.g., Simon and Perlis 2010; Kessler 2016, for reviews.

-Only 3 of the 5 authors have specific roles in the authorship statement. It is not clear that the contributions of the other 2 meet the authorship requirements of the journal.

6. PLOS authors have the option to publish the peer review history of their article (what does this mean? ). If published, this will include your full peer review and any attached files.

**Do you want your identity to be public for this peer review?** For information about this choice, including consent withdrawal, please see our Privacy Policy .

Reviewer #1: No

Reviewer #2: No

---

## [Author Response · Author response to Decision Letter 1]

17 Jul 2024

Reviewers' comments:

Reviewer's Responses to Questions

Comments to the Author

1. Is the manuscript technically sound, and do the data support the conclusions?

Reviewer #1: Yes

Reviewer #2: No

2. Has the statistical analysis been performed appropriately and rigorously?

Reviewer #1: Yes

Reviewer #2: No

3. Have the authors made all data underlying the findings in their manuscript fully available?

Reviewer #1: Yes

Reviewer #2: No

4. Is the manuscript presented in an intelligible fashion and written in standard English?

Reviewer #1: Yes

Reviewer #2: Yes

5. Review Comments to the Author

Reviewer #1: This paper investigates predictors of treatment outcome in cognitive behavioral therapy for social anxiety disorder. Potential predictors in focus are demographic factors, psychiatric history and various self-rated scales. The most predictive elements were found to be different summary scores and items on the LSAS.

The paper is interesting and well-written but it is also afflicted with a concern that needs to be addressed.

Comments:

1.This research area, prediction of treatment outcome in mental health conditions, is filled with conflicting findings and few conclusions that can be drawn despite the fact that there are plenty of studies out there. One of the reasons for this may be that researchers tend to insert too many predictors into their models, without considering from a theoretical viewpoint which predictors should be of interest to look into. This leads to chance findings and heterogenous results. I understand the potential point with the current data-driven approach, but I would like for the authors to add a paragraph in the Introduction where they discuss why the baseline predictors that they are investigating may be associated with a better or worse treatment outcome. Why could for instance initial severity of social anxiety symtoms affect the outcome? Or level of depression, sleep quality, age or education level? Based on what we know about social anxiety disorder and cognitive behavioral therapy. And the discussion should address why most predictor variables did not contribute to explain the variance in treatment outcome. What does this potentially say about CBT for social anxiety disorder?

We agree that it would be useful to have predictors with a strong conceptual basis for being likely predictors, but it is simply the state of the field, as we understand it, that besides initial symptom severity there have been no consistent predictors of treatment efficacy (response). We have added a summary of prediction papers and reviews for treatment of SAD on page 4 (including new refences). Even for symptom severity there are opposing findings whether greater or lesser initial symptom severity is associated with greater response (our finding of greater severity is consistent with single largest study). Given this, we took the most comprehensive data-driven approach possible. We also agree that it would be useful to have some insight as to why so many possible predictors did not contribute beyond the LSAS.

We have added the following sentence to the introduction:

Given the diversity of findings about what kinds of initial measures (age of onset, clinical severity, gender, and others) are associated with improvement, we included all available data that were collected from the patients in the study and used a data-driven approach.

We have also added the following two paragraphs to the discussion:

It is unclear why so many other measures did not contribute to prediction beyond the LSAS. The LSAS directly measures the symptoms of SAD, and clearly has the most face validity when considering response which is defined as a reduction of symptoms. Perhaps this is why symptom severity has been the most consistent predictor of response to CBT across studies (Mululo 2012; Rosellini 2023; 53).

The present finding that more severe pre-treatment symptoms predicted greater response differs from prior studies of SAD reporting that more severe symptoms were associated with lesser response to CBT (Mululo et al., 2012, Rosellini et al., 2023). Consistent with the present study, however, meta-analysis with data from 12 trials with 1246 SAD patients revealed that CBT led to significant symptom reduction, and individuals with higher baseline SAD severity experienced greater symptom improvement [53]. It is unclear if these differences relate to variation in patient populations, specific severity measures, or CBT delivery.

2. Only 26% of the variance was found to be explained by the 300+ predictors included in the models. What can be speculated about the 74% unexplained variance? Which are the omitted predictors?

We agree that it is a disappointingly low percent of the variance, and yet it informs the field that easily available rating scales and questionnaires account for a modest level of prediction. We note that neuroimaging studies have notably stronger predictive power, but these published studies have smaller sample sizes. Because of a data driven approach, the ML models do not omit predictors, but rather weight them as relatively uninformative. The clearest indicator of the utility of a feature, whether individually or in tandem with others, comes from the permutation analyses, which combine groups of features in all possible combinations to show that the only features relevant to performance in this dataset and across the set of models we tested where primarily the LSAS ones. Although we would, of course, have wished for a higher rate of prediction, it is useful for the aspiration of developing personalized/precision medicine that matches more patients to effective treatment to learn what measures do not work.

We have further added a statement in the discussion section regarding this as follows:

We found that the ML analyses only explained 26% of the variance in treatment response. Neuroimaging of a small subset of these patients, in combination with initial LSAS scores, explained much more of response. It is possible that the intrinsic variability of self-reported measures and the lack of biological measures (brain, genetic) and exposome measures (lifetime exposures to events prior to and during treatment) contributed to the limited variance accounted for. Analysis of self-report and clinician-report variables is the practical approach to treatment response outcome prediction without relying on hard to obtain neuroimaging data, and in a situation where other predictors simply do not exist.

3. Should the conclusion be that LSAS at baseline predicts LSAS at post-treatment?? It would be helpful with a concluding paragraph at the end of the discussion.

We have included the following paragraph at the end of the discussion. This paragraph emphasizes that baseline LSAS scores are significant predictors of post-treatment outcomes, highlighting the key findings and their implications for clinical practice. We have added following lines at the end of the discussion section

These findings do point to the value of pre-treatment LSAS scores for predicting, to a moderate degree, treatment response to CBT in SAD patients. The LSAS score is easy to acquire, and it may be a useful element in the development of a practical risk calculator for assessing the probability of individual-specific benefits of such treatment.

Reviewer #2: The authors examined the performance of 4 different modeling frameworks (ridge, lasso, support-vector regression, and extra trees) in predicting change in symptoms using up to 369 features from 157 patients who received a 12-week group CBT intervention. They observed that LASSO did the best in most comparisons and that baseline features of the symptom measure itself served as the strongest indicators of pre-post change. The best identified model explained about ~26% of the variance in the authors’ cross-validation procedure.

Developing prediction models to identify which patients are likely to respond well to particular treatments for social anxiety disorder is a laudable goal – and this paper has notable strengths. The analytic steps were generally well considered and well described, the sample was well characterized, and the manuscript was generally well written.

Some concerns dampen enthusiasm for the work in its current form:

Primary concern:

The paper seems to be aiming at two primary goals: developing a predictive algorithm and comparing different modeling algorithms. This split focus detracts, and it seems to have hindered the authors in achieving either goal. For example, in the development of the predictive algorithm, the authors only report on R-squared as the outcome. Perhaps more important for evaluating the utility of the model is information about how well (or poorly) calibrated the individual point predictions of the model are in the cross-validated sample. That is, if a clinician had access to the variables described in, e.g., table 2, what could they expect for a response profile and how confident could they be in that estimate? Regarding the second aim, it’s not clear that this data can resolve the question of which of the 4 modeling approaches is best. The sample is quite small for this sort of approach, particularly given the large number of features. There is a belief that ML techniques can accommodate large numbers of features even in small samples, but empirical tests of that hypothesis suggest it unlikely to be true (see Riley 2021 J. Clinical Epidemiology, Riley el at 2020 BMJ). Furthermore, it’s not quite clear how the results of any such comparison in a study like this wouldn’t simply be seen as sample- and context-specific, without a series of simulations to suss-out more general features that could help future researchers determine which approach is likely to be better for their sample and context. Furthermore, there have been advancements in ensemble-based ML methods that attempt to sidestep entirely needing to choose the one “best” algorithm for a particular application.

We thank the reviewer for their comment.

The primary focus of the work was to evaluate if a set of pre-treatment clinical and behavioral assessment information could predict treatment outcome. If this worked out, which it did, the secondary aim was to determine which components of such assessments are important. Predictive models are used as a tool that has two key benefits. First, it addresses both these questions simultaneously. Second, and perhaps most importantly, it ensures that we always evaluate out-of-sample performance through cross-validation, and thus such models can be applied to unseen data.

Multiple models were used to ensure that the mathematical peculiarities of each model did not bias our interpretation of relevant and important features. Our primary intent was to understand contributors and not to optimize performance beyond individual models. The distributions in the results are a measure of confidence for each model. These specific models do not provide a measure of individual level uncertainty.

All models were evaluated through bootstrapped cross-validation, that is, each train-test split was randomly sampled from the data. We believe this is in concordance with the methods described in the Riley papers. Further, in the permutation analyses (see Fig 3) the ratio of participants to number of features vary in each analysis and the clearest outcome is still the improvement in performance when including the LSAS assessment, independent of the total number of features.

We have added following lines:

The primary aim of this research was to establish if treatment response outcome can be predicted for CBT in SAD patients using readily available information. We used an out-of-sample predictive modeling approach with bootstrapped train-test splits to answer this question. As models can vary in their calculation of feature weights, we also compared the performance of a few models to better understand which features are relevant to the problem.

We also added in limitations that:

One of the limitations of this study was the reliance on data from a clinical trial that has a limited sample size in relation to the many individual and environmental factors that might influence outcome.

Additional concerns:

-Given the sophistication of the rest of the analyses, I was surprised by the choice of imputation strategy (mode imputation) and the lack of detail provided regarding missing data. How much data needed to be imputed, both for the primary outcome and each of the IVs? Although straightforward, mode imputation does not preserve any of the relationships between the variables. Additional detail and justification is required.

We have provided detailed information on the amount of missing data and justified our choice of mode imputation. We have also considered and discussed alternative imputation strategies that preserve relationships between variables, ensuring that our approach is robust and justified.

We have added following in the limitation section:

Our data has 10% missing values across different variables. One of the challenges in selecting the best imputation strategy was that we did not know a priori if there exists dependencies between certain variables. All the missing variables were from the categorical variables and therefore mode imputation is most suited for such kind of missing values. When data were restricted to a subset of participants who had no missing data, the analyses (not included) yielded near identical results and the same conclusions.

-Additional discussion of the limitations surrounding the lack of a comparison group are warranted. The fact that baseline values were the strongest predictor suggests the real possibility of regression-to-the-mean confounding the results. Moreover, without a comparison, care needs to be taken in characterizing how such models can be used. For example, the model may predict that Patient X will fare poorly in this treatment, but it is possible that they would fare even worse in other treatments.

We discuss the limitations of our study due to the lack of a comparison group that was not present in the trial dataset. We acknowledged the possibility of regression-to-the-mean and clarified how this limitation affects the interpretation of our predictive models. This discussion emphasizes the careful application of our models in clinical settings. We have added the following in the discussion section

Lack of a comparison group is a limitation of this dataset and this study. A comparison treatment, such as a pharmacological treatment, might help reveal how alternative treatments should be weighed for a patient. This would reflect the practical clinical need to offer the treatment most likely to help a patient, even if that treatment has a modest likelihood of being he

---

## [Decision Letter · Decision Letter 1]

20 Aug 2024

PONE-D-23-36553R1Predicting Treatment Response to Cognitive Behavior Therapy in Social Anxiety Disorder on the Basis of Demographics, Psychiatric History, and Scales: A Machine Learning ApproachPLOS ONE

Dear Dr. Bukhari,

Thank you for submitting your manuscript to PLOS ONE. After careful consideration, we feel that it has merit but does not fully meet PLOS ONE’s publication criteria as it currently stands. Therefore, we invite you to submit a revised version of the manuscript that addresses the points raised during the review process.

 Please address Reviewer 2's outstanding comment. 

We look forward to receiving your revised manuscript.

Kind regards,

Kymberly D. Young, Ph.D.

Academic Editor

PLOS ONE

Journal Requirements:

Reviewers' comments:

Reviewer's Responses to Questions

**Comments to the Author**

1. If the authors have adequately addressed your comments raised in a previous round of review and you feel that this manuscript is now acceptable for publication, you may indicate that here to bypass the “Comments to the Author” section, enter your conflict of interest statement in the “Confidential to Editor” section, and submit your "Accept" recommendation.

Reviewer #1: All comments have been addressed

Reviewer #2: (No Response)

2. Is the manuscript technically sound, and do the data support the conclusions?

Reviewer #1: Yes

Reviewer #2: Yes

3. Has the statistical analysis been performed appropriately and rigorously? 

Reviewer #1: Yes

Reviewer #2: Yes

4. Have the authors made all data underlying the findings in their manuscript fully available?

Reviewer #1: No

Reviewer #2: No

5. Is the manuscript presented in an intelligible fashion and written in standard English?

Reviewer #1: Yes

Reviewer #2: Yes

6. Review Comments to the Author

Reviewer #1: I believe the authors have addressed and responded well to all the critique provided by the reviewers, to the best of their capacity and given the limitations of the study.

Reviewer #2: The authors have largely been responsive and most prior concerns have been addressed. The one lingering issue is that it's not clear how clinicians, or researchers, are meant to use these findings moving forward. The authors highlight that adding subscores and individual items to a model containing the total LSAS scores improves prediction. It would be helpful if the authors could, in plain language, explain how someone who wanted to make use of these findings should go about doing so.

7. PLOS authors have the option to publish the peer review history of their article (what does this mean? ). If published, this will include your full peer review and any attached files.

**Do you want your identity to be public for this peer review?** For information about this choice, including consent withdrawal, please see our Privacy Policy .

Reviewer #1: No

Reviewer #2: No

---

## [Author Response · Author response to Decision Letter 2]

21 Oct 2024

Dear Reviewers

Thank you for considering our manuscript. We have made changes as requested. Please find our updated manuscript with the change as requested.

Specifically, we have added following lines in order to address the request of the reviewer.

"Clinicians and researchers typically use overall summary scores from self-report questionnaires to evaluate diagnoses or response to treatment. Indeed, such summary scores typically have the most reliability and validity relative to subscores or individual items. In the present study, however, prediction was most accurate when data from individual items, subscores, and the total score were all included. A similar benefit of including all data from a single clinical questionnaire for ML prediction was observed for a study predicting the development of bipolar disorder in children over a 10-year period [56]. At a practical level, clinicians and researchers would not be expected to implement ML algorithms to calculate likelihood of treatment efficacy (or development of a diagnosis), but rather risk calculators would need to be made available that would automatically compute such probabilistic outcomes. Such risk calculators are already in use in other areas of medicine, but not yet for psychiatric diagnoses or treatments."

Thank you

---

## [Editor Report · Decision Letter 2]

23 Oct 2024

Predicting Treatment Response to Cognitive Behavior Therapy in Social Anxiety Disorder on the Basis of Demographics, Psychiatric History, and Scales: A Machine Learning Approach

PONE-D-23-36553R2

Dear Dr. Bukhari,

We’re pleased to inform you that your manuscript has been judged scientifically suitable for publication and will be formally accepted for publication once it meets all outstanding technical requirements.

Kind regards,

Kymberly D. Young, Ph.D.

Academic Editor

PLOS ONE

---

## [Editor Report · Acceptance letter]

PONE-D-23-36553R2

PLOS ONE

Dear Dr. Bukhari,

I'm pleased to inform you that your manuscript has been deemed suitable for publication in PLOS ONE. Congratulations! Your manuscript is now being handed over to our production team.

Kind regards,

on behalf of

Dr. Kymberly D. Young

Academic Editor

PLOS ONE